# COVID-19-related social support service closures and mental well-being in older adults and those affected by dementia: a UK longitudinal survey

Clarissa Giebel [1,2] Daniel Pulford,[3] Claudia Cooper,[4] Kathryn Lord,[5] Justine Shenton,[6] Jacqueline Cannon,[7] Lisa Shaw,[8] Hilary Tetlow,[9] Stan Limbert,[2] Steve Callaghan,[10] Rosie Whittington,[11] Carol Rogers,[12] Aravind Komuravelli,[13] Manoj Rajagopal,[3] Ruth Eley,[14] Murna Downs,[15] Siobhan Reilly,[16] Kym Ward,[17] Anna Gaughan,[18] Sarah Butchard,[1] Jules Beresford,[5] Caroline Watkins,[19] Kate Bennett,[1] Mark Gabbay[1,2]

For numbered affiliations see end of article.

**Correspondence to**
Dr Clarissa Giebel;
clarissa.giebel@liverpool.ac.uk

## ABSTRACT

**Background** The COVID-19 pandemic has had a major impact on delivery of social support services. This might be expected to particularly affect older adults and people living with dementia (PLWD), and to reduce their well-being.

**Aims** To explore how social support service use by older adults, carers and PLWD, and their mental well-being changed over the first 3 months since the pandemic outbreak.

**Methods** Unpaid dementia carers, PLWD and older adults took part in a longitudinal online or telephone survey collected between April and May 2020, and at two subsequent timepoints 6 and 12 weeks after baseline. Participants were asked about their social support service usage in a typical week prior to the pandemic (at baseline), and in the past week at each of the three timepoints. They also completed measures of levels of depression, anxiety and mental well-being.

**Results** 377 participants had complete data at all three timepoints. Social support service usage dropped shortly after lockdown measures were imposed at timepoint 1 (T1), to then increase again by T3. The access to paid care was least affected by COVID-19. Cases of anxiety dropped significantly across the study period, while cases of depression rose. Well-being increased significantly for older adults and PLWD from T1 to T3.

**Conclusions** Access to social support services has been significantly affected by the pandemic, which is starting to recover slowly. With mental well-being differently affected across groups, support needs to be put in place to maintain better well-being across those vulnerable groups during the ongoing pandemic.

## INTRODUCTION

In the UK, 11.9 million people are aged 65 and over,[1] with over 850 000 living with dementia.[2] Social support services, including day care centres, support groups, paid home carers and community activities, such as singing or arts groups, are important for maintaining a good quality of life for older people and people living with dementia (PLWD).[3 4] In view of an ageing population and increasing numbers of PLWD, easily accessible services are crucial to support people socially, as well as with their care needs.

The COVID-19 pandemic has affected these social support services significantly. Social isolation as a result of social distancing, lockdowns and shielding is a huge concern for older people across the globe since the beginning of the pandemic,[5 6] with similar issues highlighted early for PLWD.[7] However, there is still a dearth of evidence on the mental well-being and access to care for those in need.

Specifically, in the UK, a nationwide 3-month lockdown was imposed on 23rd of March. Older people were over-represented in the group who were clinically extremely

vulnerable, that is at greatest risk of severe illness from COVID-19, who were asked to shield by the government until early August. All adults aged 70+ were classified as being at least moderate risk of severe illness from COVID-19.[8] During the most restrictive, earlier period of lockdown, people were advised to only go outside once a day for essential food shopping, pharmacy visits, or to exercise. Non-essential shops were closed, and only started reopening in July. With additional social distancing for the general population, and use of personal protective equipment (PPE) for the health and social care taskforce in place, these measures significantly impact the social support services that PLWD, carers and older adults could receive. Recent qualitative evidence has highlighted how PLWD and unpaid carers have faced a sudden crisis in terms of accessing social support services since the pandemic,[9] and have faced difficult decisions whether to continue or discontinue paid carers entering the home of the PLWD, for fear of potential virus transmission.[10] While these qualitative accounts provide rich information on the experiences of having accessed (or failed to access) social support services during the pandemic, there appears to be no empirical evidence to date quantifying those experiences and linking these with mental well-being.

## Strengths and limitations of this study

► Data on social support service usage and mental health were collected at three timepoints in the early stages of the pandemic.
► Sixty-three per cent of participants completed all three survey timepoints.
► The survey was coproduced with people affected by dementia.
► The study was mostly completed by people from a White ethnic background, and lacks ethnic minority representation despite actively approaching community groups.
► Some people with dementia completed the survey, who may have had difficulties in recalling the services they received pre pandemic.

The aim of this exploratory study was to explore the impacts of COVID-19 on social support service closures and longitudinal changes in the mental illness and well-being of older adults, PLWD and unpaid carers. Considering the new emergence of this field and thus lack of previous evidence, we hypothesised that would be associated with reduced social support service provision, which in turn was hypothesised to be associated with poorer mental health.

## METHODS
### Participants and recruitment
We recruited UK residents who were aged 18+. PLWD were eligible to take part if they had a diagnosis of dementia. Unpaid carers were eligible to take part if they were (current carers) or had been caring for a relative or friend with dementia (former carers). Older adults were eligible to take part if they were aged 65 years or older.

Participants were recruited via different social support services third sector organisations, such as peer support group organisations, carer networks, cultural dementia training programme organisations and national dementia subtype specific organisations, and by contacting people on their email circulation lists, via newsletters and social media accounts. We also directly contacted people who were accessing regular services, such as support groups or older people fora, via telephone. This ensured that people without internet access were able to participate in this research. We also used Join Dementia Research, a UK-wide national online register of PLWD, carers, older adults and health volunteers who are interested in taking part in dementia and ageing research.

### Data collection
The study was completed at three timepoints (T1, T2 and T3), 5 and 6 weeks apart, respectively. Participants could complete the survey either online or over the phone with a research team member who entered their details into the online survey on their behalf. Participants from T1 (baseline) were followed-up with the same mental well-being questionnaires at T2 and T3 and were followed-up either by telephone or email, depending on how they completed T1 survey. T1 ran from 17 April to 15 May (±3 days). T2 ran from 29 May to 26 June (±3 days). T3 ran from 10 July to 7 August (±3 days).

### Variables and tools
At T1, participants were asked about their background characteristics (including age, gender, ethnicity, postcode, living situation, type of dementia (if applicable) and employment). Postcode data were collected to generate an Index of Multiple Deprivation (IMD) quintile. IMD provides a measure of neighbourhood deprivation, taking into account income, education, crime and health, among others. Quintile 1 indicates least deprived neighbourhoods, with quintile 5 indicating the most deprived neighbourhoods.

Service usage was measured by asking about pre pandemic and current receipt of different social support services (including paid carers, support groups, befrienders, day care centres, respite, meal deliveries, transport, social activities, clinical mental health support and clinical physical support) and equipment, such as hand rails or shower seats, as well as the weekly total hours of social support services. Prepandemic service usage was defined as use of social support services in a typical week before the pandemic.

Participants were also asked to complete the Personalised Health Questionnaire 9 (PHQ-9)[11] for levels of depressive symptoms, the Generalised Anxiety Disorder 7 (GAD-7)[12] for levels of anxiety symptoms, and the Short Warwick-Edinburgh Mental Well-Being Scale[13] (SWEMWBS) for quality of life. Higher scores on the PHQ-9, GAD-7 and the SWEMWBS indicated higher levels of depressive symptomatology, anxiety symptomatology and quality of life, respectively. We categorised participants who scored of 10 or more on the PHQ-9 as 'depressed' and on the GAD-7 as 'anxious', as based on previous extensive research, indicating that these cut offs indicate general anxiety disorder and depression, respectively.[14] At T2 and T3, participants were asked again about their current levels of social support service receipt, weekly hours of support, equipment, as well as the PHQ-9, GAD-7 and SWEMWBS.

### Data analysis
Data were analysed using SPSS V.25, and the significance level was set at $p<0.05$. Participant demographic characteristics and social support service usage and mental well-being variables were analysed using frequency analysis. $\chi^2$ tests were used to assess variations in the proportions

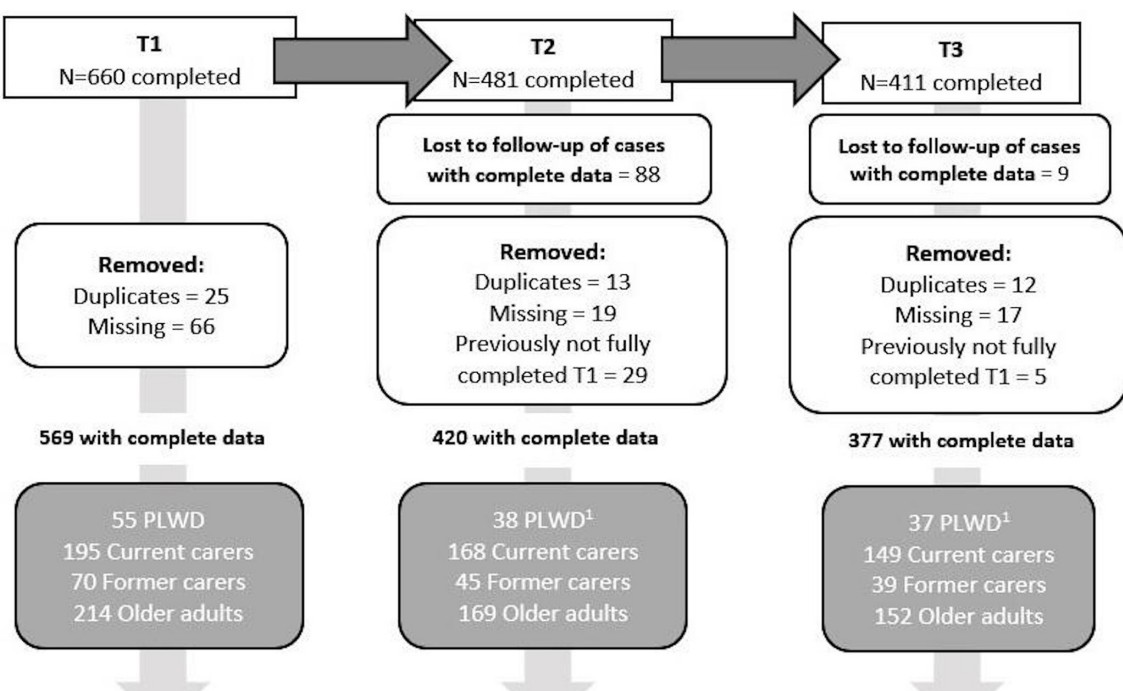

**Figure 1** Flow of participation in longitudinal survey. The top boxes indicate how many people completed each survey timepoint. After having removed (1) duplicates (people who completed the survey two times), (2) missing cases (where participants had not completed the Personalised Health Questionnaire 9, Generalised Anxiety Disorder 7, and the Short Warwick-Edinburgh Mental Well-Being Scale or had missing ID codes at T2 and T3) and (3) losses to follow-up (those that had either completed T1 or T1 and T2 only), and (4) incomplete data at T1 yet data at T2 or T3, 377 cases remained in total. Grey boxes indicate the breakdown by subgroup. [1] Follow-up completion by subgroup by percentage at T2 and T3 compared with T1: people living with dementia (PLWD) 69% (T2), 67% (T3); Current carers 86% (T2), 75% (T3); Former carers 64% (T2), 60% (T3); Older adults 79% (T2), 69% (T3).

of participants who were categorised as 'depressed' or 'anxious'. Repeated measures Analysis of Variances (ANOVAs) with Greenhouse Geisser posthoc correction were used to analyse differences between T1, T2 and T3 in GAD-7 total, PHQ-9 total and SWEMWBS total scores. For this analysis, only participants with complete GAD-7, PHQ-9 and SWEMWBS were included (n=377). Paired samples t-tests were employed to compare the means of GAD-7, PHQ-9 and SWEMWBS at T1 between those who completed all three timepoints (n=377) and those who dropped out after T1 or T2 (n=192). Bivariate correlation analysis was employed to assess whether changes in weekly social support service usage from pre pandemic to T1 were associated with changes in GAD-7, PHQ-9 and SWEMWBS between T1 and T3.

### Patient and public involvement
Unpaid carers and a PLWD were involved as equal team members in all aspects of the study—from conceptualisation and design through to analysis and dissemination.

## RESULTS
### Survey completion
Figure 1 outlines the participant flow and completion rates in further detail. Overall, 569 participants

completed the survey at T1 (61 PLWD; 219 current carers; 66 former carers; 223 older adults). Overall, 420 participants completed the survey at T2 (38 PLWD; 168 current carers; 45 former carers; 169 older adults). Overall, 377 participants completed all three waves of the survey (37 PLWD; 149 current carers; 39 former carers; 152 older adults).

### Participant characteristics
Table 1 shows the demographic characteristics of those who completed T1 and those that completed all three survey timepoints, by subgroup. For those who completed all three timepoints, carers and older adults were mostly female (59%–82%), while the majority of PLWD were male (62%). The majority of participants were from a White ethnic background (95%–99%) and lived with someone else (61%–88%), with current carers having the highest proportion of living with someone else. The majority of participants across all four groups lived in less deprived neighbourhoods (quintiles 1 and 2) (52%–61%). Thirty-seven PLWD took part in all three timepoints; the most common diagnostic subtype was Alzheimer's disease.

### Social support service and activities usage
Participants had accessed a range of social support services pre pandemic, including day care centres, support groups, meal deliveries, respite and paid carers.

**Table 1** Participant characteristics of those completing T1 survey and those completing all three survey timepoints

| | T1 (n=569) | | | | T1, T2 and T3 (n=377) | | | |
|---|---|---|---|---|---|---|---|---|
| | PLWD (n=61) | Current carers (n=219) | Former carers (n=66) | Older adults (n=223) | PLWD (n=37) | Current carers (n=149) | Former carers (n=39) | Older adults (n=152) |
| **N (%)** | | | | | | | | |
| Gender | | | | | | | | |
| Female | 27 (44.3) | 168 (77.1) | 55 (83.3) | 137 (61.7) | 14 (37.8) | 118 (79.7) | 32 (82.1) | 90 (59.2) |
| Male | 34 (55.7) | 50 (22.9) | 11 (16.7) | 85 (38.3) | 23 (62.2) | 30 (20.3) | 7 (17.9) | 62 (40.8) |
| Ethnicity | | | | | | | | |
| White | 58 (96.7) | 211 (96.3) | 65 (98.5) | 216 (98.2) | 35 (94.6) | 143 (96.0) | 38 (97.4) | 148 (98.7) |
| Other | 2 (3.4) | 8 (3.7) | 1 (1.5) | 4 (1.9) | 2 (5.4) | 6 (4.0) | 1 (2.6) | 2 (1.3) |
| Living situation | | | | | | | | |
| Alone | 13 (21.3) | 33 (15.1) | 17 (26.2) | 79 (35.6) | 8 (21.6) | 18 (12.2) | 11 (28.9) | 59 (39.1) |
| With someone | 48 (78.7) | 185 (84.9) | 48 (73.8) | 143 (64.4) | 29 (78.4) | 130 (87.8) | 27 (71.1) | 92 (60.9) |
| Index of Multiple Deprivation Quintile | | | | | | | | |
| 1 | 12 (23.1) | 54 (32.1) | 10 (19.2) | 61 (33.5) | 5 (16.1) | 35 (31.0) | 6 (19.4) | 39 (31.7) |
| 2 | 16 (30.8) | 50 (29.8) | 20 (38.5) | 44 (24.2) | 11 (35.5) | 34 (30.1) | 11 (35.5) | 29 (23.6) |
| 3 | 10 (19.2) | 32 (19.0) | 14 (26.9) | 37 (20.3) | 6 (19.4) | 20 (17.7) | 9 (29.0) | 26 (21.1) |
| 4 | 10 (19.2) | 14 (8.3) | 5 (9.6) | 26 (14.3) | 5 (16.1) | 11 (9.7) | 4 (12.9) | 18 (14.6) |
| 5 | 4 (7.7) | 18 (10.7) | 3 (5.8) | 14 (7.7) | 4 (12.9) | 13 (11.5) | 1 (3.2) | 11 (8.9) |
| Type of dementia | | | | | | | | |
| Alzheimer's | 20 (32.8) | 100 (46.5) | 6 (23.1) | | 14 (37.8) | 75 (50.7) | 2 (20.0) | |
| Mixed | 13 (21.3) | 49 (22.8) | 7 (26.9) | | 6 (16.2) | 34 (23.0) | 4 (40.0) | |
| Vascular | 11 (18.0) | 27 (12.6) | 4 (15.4) | | 8 (21.6) | 18 (12.2) | 3 (30.0) | |
| Other | 17 (27.9) | 39 (18.1) | 9 (34.5) | | 9 (24.3) | 21 (14.1) | 1 (10.0) | |
| **Mean (SD), (range)** | | | | | | | | |
| Age | 70 (±10), (45–88) | 61 (±13), (23–89) | 64 (±14), (22–95) | 72 (±6), (65–90) | 72 (±10), (50–88) | 62 (±13), (23–89) | 65 (±13), (22–95) | 73 (±6), (65–90) |
| Years of education | 15 (±4), (4–25) | 16 (±4), (6–28) | 17 (±4), (10–29) | 17 (±4), (7–25) | 13 (±4), (4–20) | 16 (±4), (6–28) | 16 (±4), (10–29) | 16 (±4), (7–24) |

Five hundred and sixty-nine participants completed the survey at T1, with duplicates and missing cases removed. Three hundred and seventy-seven participants had completed all three survey timepoints, with duplicates and missing cases removed.
PLWD, People living with dementia.

Figure 2A shows the proportion of participants of the total sample (n=377) who reported accessing paid carers, support groups, day care, befrienders and social activities prior to the pandemic, and at T1, T2 and T3. These were the most commonly used types of social support services prior to the pandemic. Social support services usage had dropped since the pandemic outbreak. Pre pandemic, 27% of participants accessed social activities in the community, which dropped to 6% at T1, T2 and T3. Paid care saw the smallest change—with 17% having accessed paid carers pre pandemic, dropping to 12% at T1 and increasing slightly again to 15% at T3. Day care saw the largest drop, with only 1%–2% receiving day care since the outbreak, compared with 15% previously.

Figure 2B shows the proportion of participants by group who have received any form of social support services pre pandemic and at all three survey timepoints. Pre pandemic, 90% of current carers had received social support of any form, with between 45% and 50% of former carers and older adults having received some

support. This decreased at T1 for all groups to between 20% (older adults) and 55% (current carers) receiving some type of support. Through T2 and T3, an upward trend emerged with more participants gaining access to some services again, with levels for PLWD and former carers being higher at T3 than at prepandemic levels.

### Mental well-being

Figure 3 shows the proportion of participants across the total sample (n=377) who were categorised and identified as anxious and depressed, based on scoring above the GAD-7 and PHQ-9 cut-off, across all three timepoints. For anxiety, we noted a downward trend in number of cases from T1 (16.5%) to T3 (14.1%). The proportion of participants with anxiety was significantly lower at T2 ($x^2=186.399$, p<0.001) and T3 ($x^2=136.562$, p<0.001) compared with T1. For depression, we noted an upward trend in cases, as indicated based on their cut-off on the PHQ-9, from T1 (14.4%) to T3 (17.5%). The proportion of participants with depression was significantly higher at

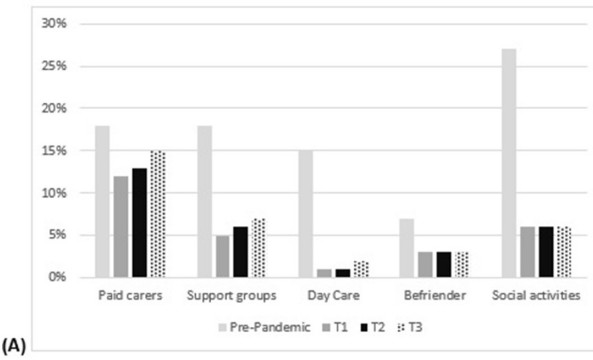

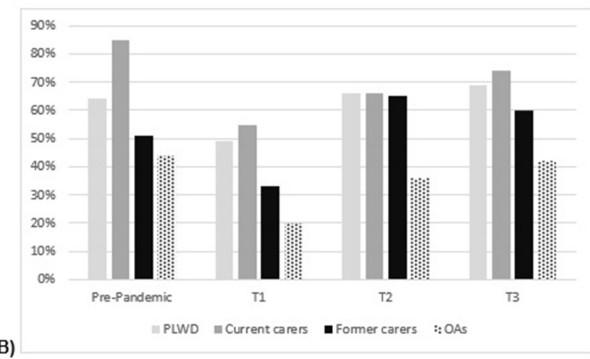

**Figure 2** Social support service usage pre pandemic and at three survey timeoints. (A) Service usage for the total sample (N=377) in proportion of participants at four different timepoints for some of the most frequently used support services. (B) Proportion of participants within each group at four different timepoints (pre pandemic, T1, T2 and T3) having received any form of social support.

T2 ($x^2$=176.248, p<0.001) and T3 ($x^2$=158.031, p<0.001) compared with T1.

Figure 4 shows the median of the GAD-7, PHQ-9 and SWEMWBS total scores by group over time for those who completed all three survey timepoints. Based on the median scores, levels of anxiety and depression appear

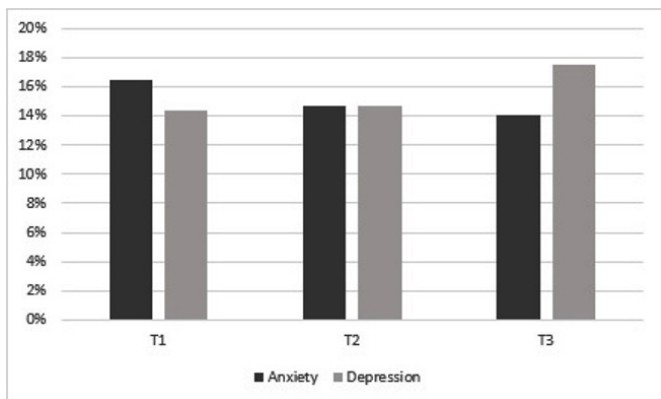

**Figure 3** Proportion of the total sample who scored above the cut offs for anxiety and depression at three timepoints. T=Timepoin. The graph shows the proportion of participants from the total sample who completed all three surveys (n=377) and scored above the cut-off on the Generalised Anxiety Disorder 7 and Personalised Health Questionnaire 9 for anxiety and depression, respectively.

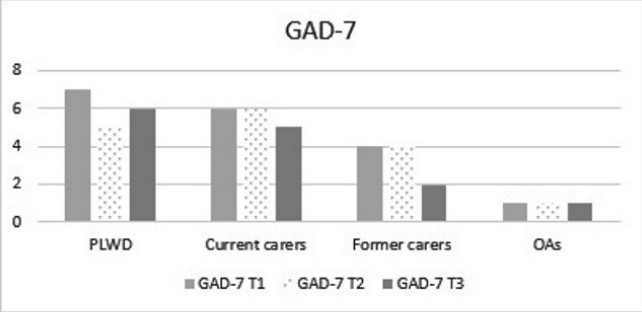

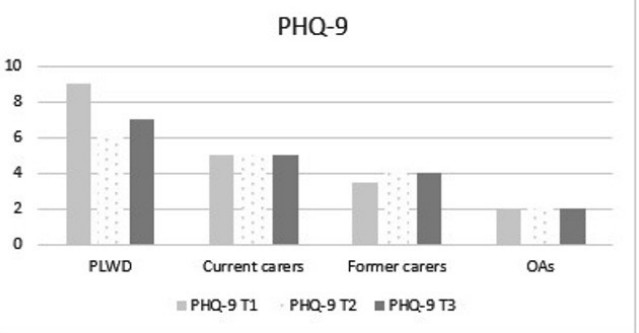

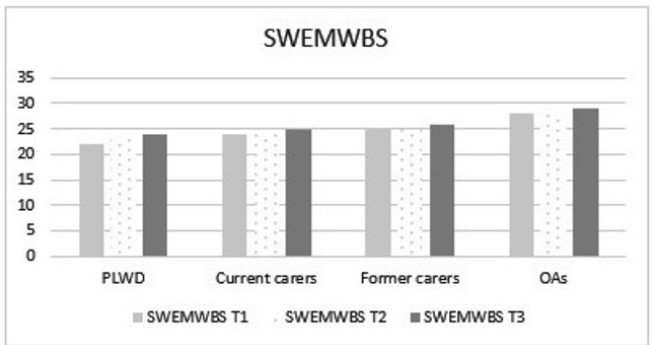

**Figure 4** Variations in anxiety, depression and quality of life total scores at three timepoints. Figures show the median total score at each timepoint (T1, T2 and T3) for each subgroup—for anxiety (GAD-7), depression (PHQ-9) and quality of life (SWEMWBS). AOs, Older adults; GAD-7, Generalised Anxiety Disorder 7; PHQ-P, Personalised Health Questionnaire 9; PLWD, people living with dementia; SWEMWBS, Short Warwick-Edinburgh Mental Well-Being Scale.

to decrease from T1 to T3, while quality of life increases from T1 to T3.

Paired samples t-test showed that there were no significant differences in means of GAD-7 (p=0.468), PHQ-9 (p=0.183) and SWEMWBS (p=0.332) at T1 between those who completed all three survey timepoints and those who dropped out after T1 or T2.

For anxiety, repeated measures ANOVAs with Greenhouse-Geisser posthoc correction showed that GAD-7 total scores did not vary significantly from T1 to T3 for PLWD ($F$(1.856, 64.962)=1.429, p=0.247) or among current carers ($F$(1.898, 277.063)=1.938, p=0.148], former carers ($F$(1.801, 68.419)=0.139, p=0.139), or older adults ($F$(1.924, 286.727)=2.688, p=0.0072), based

on those participants who completed in all three survey timepoints.

For depression, repeated measures ANOVAs showed that PHQ-9 total scores did not significantly vary from T1 to T3 for PLWD ($F(1.896, 66.370)=1.461$, $p=0.240$), current carers ($F(1.900, 277.453)=0.639$, $p=0.521$), former carers ($F(1.677, 68.419)=0.024$, $p=0.960$), or older adults ($F(1.889, 281.414)=0.857$, $p=0.420$).

For well-being, repeated measures ANOVAs showed that SWEMWBS total scores significantly increased from T1 to T3 for PLWD ($F(1.726, 60.423)=5.412$, $p<0.05$ (Mean (SD) T1–T3=22.1 (6.4); 24.3 (5.6); 24.4 (5.6))) and older adults ($F(1.804, 268.807)=3.632$, $p<0.05$ (Mean (SD) T1–T3=27.5 (5.1); 28.0 (5.1); 28.3 (4.9))). However there were no significant changes among current ($F(1.982, 289.325)=2.185$, $p=0.115$) or former carers ($F(1.728, 63.725)=0.268$, $p=0.733$).

### Social support and mental health

Bivariate correlation analyses showed no significant associations between variations in social support service hours between pre pandemic and at T1 and changes in SWEMWBS ($p=0.332$), GAD-7 ($p=0.310$), and PHQ-9 ($p=0.351$) between T1 and T3 for those who completed all three survey timepoints.

### DISCUSSION

This is one of the first studies to show that social support service usage in dementia and ageing reduced significantly compared with prepandemic levels, while slowly rising in the months post nationwide lockdown. In addition, we also show that cases of anxiety reduced while cases of depression increased in the months since lockdown, with quality of life significantly increasing for PLWD and older adults only.

Social support service usage for PLWD, unpaid carers, and older adults has seen a significant decrease since the COVID-19 pandemic, leaving many people suddenly without vital support—ranging from day care centres to respite to support groups. In the months following the nationwide lockdown, usage has gradually increased again but varied among providers and type of support. With public health restrictions still remaining in place, however, during that period, including social distancing, shielding and thus inability to meet members of different households, such support is most likely to be implemented via digital technologies. Considering that in our sample 94% of participants completed the survey online rather than the telephone option, nearly all participants had access to the internet. However, many older adults and PLWD are less likely to be digitally literate,[15] making it difficult for all people to access services equally. This has already been an issue pre COVID-19,[16] suggesting that the pandemic has further exacerbated potential inequalities in access and thus further isolated people who would benefit from social support the most.

One type of support which has been affected the least by the pandemic has been paid home care. Receiving paid home care enables PLWD and older adults to stay at home independently for longer—as people wish to avoid entering a care home and stay in their familiar environment.[17] While there was a reduction in paid home care usage compared with prepandemic levels, overall paid home carers were used the most. A qualitative exploration into decision-making for whether or not to continue paid home care during the pandemic has shown that many unpaid carers were afraid of having paid carers enter the home (often with inadequate PPE) for risk of potential virus transmission.[10] Other unpaid carers however felt unable to cope without the support, or indeed accepted the potential risks, and continued paid home care. There is also a notable difference between social care provision (which is paid home care) and third sector care provision (which involves support groups and social activities for example). The third sector relies on volunteers providing services, and has also suffered during the pandemic, whereas the social care sector is financially supported by the government. Therefore, the ability to receive home care might not have been affected to the same extent as accessing support groups for instance. Another potential reason for variations in usage between activity types is that home care involves someone from the outside entering someone's home. In contrast, day care centres, respite care and social activities involve older adults and PLWD going outside to larger social gatherings. Due to public health restrictions, these have been temporarily rendered largely, if not completely, impossible to take place in their original face-to-face formats. As numbers of infections rise again, these restrictions are being strengthened and reimposed with large fines possible for those transgressing them.

Levels of anxiety, depression and well-being changed over the course of the study period. Over 12 weeks, cases of anxiety across the total sample dropped, while cases of depression increased significantly. However, when exploring levels of anxiety and depression within groups, no significant changes were noted, which is likely to have been due to small and varied sample sizes for each subgroup. Similarly, no significant variations in levels of anxiety, depression and mental well-being were found between those who completed all three survey timepoints and those who had dropped out after T1 or T2. It is possible that participants felt more connected over time, particularly considering again that the majority of participants completed the survey online and thus were able to participate in remote services, where these existed. Recent evidence from Spain showed how older adults were less likely to suffer from psychological distress as a result of the pandemic than people aged below 60.[18] Nevertheless, overall the pandemic is having a heightened impact on the mental health of the general population.[19 20] Engaging in social activities can be one avenue to help maintain good mental health.[21] Considering that reductions in social engagement both before and after

a dementia diagnosis are common,[22] enabling continued engagement throughout the pandemic is important to support PLWD, carers and older adults adequately. This is corroborated by evidence from the baseline survey showing that reductions in social support usage were linked to mental well-being.[23] It is possible that for this study, merging groups of older adults, PLWD and carers resulted in no significant associations, as each group was differently affected, as indicated by looking at changes of mental well-being for each group across the 12 weeks.

There were some limitations to our exploratory study. While benefitting from a large sample size and good retention rate over a relatively short time period of 12 weeks, there was some missing data and not everyone completed all three survey timepoints. However, this is standard in longitudinal survey-based research, and we still generated a large sample size across all three timepoints. By comparing those who completed all three survey timepoints and those who dropped out after T1 or T2, we established that there no significant differences in their mental health scores. Concerning the participant population, it is to be noted that the majority of participants had internet access and were thus also able to join in remote social support. Although we actively approached older adults, PLWD and carers via phone through recruiting organisations, only some people took part over the phone. It is likely, however, that those people without internet access have been even more isolated through the pandemic, with potentially severe mental health needs, which we have only captured a snapshot of. This also links to the fact that there are likely to be longer-term effects on mental well-being, with our survey only providing a snap shot of the first few months since the start of the pandemic. Equally, our survey did not include prepandemic levels of mental well-being (anxiety, depression and quality of life), which would have provided additional insight into changes in mental well-being. However, due to the unforeseen circumstance of the pandemic, it was not feasible to collect these data. We only enquired about weekly hours of total social support usage, and not for each specific type of activity. Some participants might have accessed, for example, paid home care, but only for 2 hour as opposed to others who might have received 40 hours a week. We are thus unable to state in detail how the pandemic has affected the level of each different type of support, but instead we provide a more general overview of activities and general service usage variations since the nationwide lockdown, which to existing knowledge has not been captured elsewhere.

## CONCLUSIONS

The pandemic is having a sudden and severe long-term impact on social support service usage for older adults and people affected by dementia, which sees somewhat of a limited increase in usage over the first few months since nationwide lockdown. While it appears that some services have started providing remote support, not everyone will be able to access these, leaving many people without much needed support. Future research needs to assess how older adults and people affected by dementia are accessing social support services in the time of COVID-19, with clearer support for people to access any format of services—either face to face or remotely. Considering that the pandemic is going to continue for the foreseeable future, the mental health of older adults and those affected by dementia needs to be closely monitored, particularly when more stringent public health measures are put in place again.

**Author affiliations**
[1]Institute of Psychology, Health and Society, University of Liverpool, Liverpool, UK
[2]NIHR ARC NWC, Liverpool, UK
[3]Lancashire Care NHS Foundation Trust, Preston, UK
[4]University College London, London, UK
[5]University of Bradford, Bradford, UK
[6]Sefton Older People's Forum, Sefton, UK
[7]Lewy Body Society, Wigan, UK
[8]Department of Modern Languages and Cultures, University of Liverpool, Liverpool, UK
[9]SURF Liverpool, Liverpool, UK
[10]EQE Health, Liverpool, UK
[11]Me2U Care, Liverpool, UK
[12]National Museums Liverpool, Liverpool, UK
[13]North West Boroughs Healthcare NHS Foundation Trust, Warrington, UK
[14]Liverpool DAA, Liverpool, UK
[15]Bradford Dementia Group, University of Bradford, Bradford, UK
[16]Division of Health Research, Lancaster University, Lancaster, UK
[17]The Brain Charity, Liverpool, UK
[18]TIDE, Liverpool, UK
[19]Faculty of Health and Wellbeing, University of Central Lancashire, Preston, UK

**Acknowledgements** This study would not have been possible without our participants, thank you very much. We wish to thank Join Dementia Research for helping us recruit to our study. We also wish to thank Emma Riley who helped with recruitment by emailing eligible participants via the House of Memories networks.

**Contributors** CG managed data collection, conducted analysis and wrote the manuscript. JC, JS and DP collected data over the phone. DP managed the data. All authors contributed to designing the survey, interpreting the findings and reading drafts of the manuscript. All authors approved the final manuscript.

**Funding** This research is supported by a grant awarded to the authors by the University of Liverpool COVID-19 Strategic Research Fund in 2020. This is also independent research funded by the National Institute for Health Research Applied Research Collaboration North West Coast. The views expressed in this publication are those of the author(s) and not necessarily those of the National Institute for Health Research or the Department of Health and Social Care. The University of Bradford QR Research Fund also supported part of this study. There are no grant numbers.

**Competing interests** None declared.

**Patient and public involvement** Patients and/or the public were involved in the design, or conduct, or reporting, or dissemination plans of this research. Refer to the Methods section for further details.

**Patient consent for publication** Not required.

**Ethics approval** Ethical approval was obtained from the University of Liverpool prior to study begin (Ref: 7626).

**Provenance and peer review** Not commissioned; externally peer reviewed.

**Data availability statement** Data may be obtained from a third party and are not publicly available. Data may be obtained by a third party and are not publicly available.

**ORCID iD**
Clarissa Giebel http://orcid.org/0000-0002-0746-0566

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
