## [Reviewer comments · BMJ Open]

ARTICLE DETAILS

TITLE (PROVISIONAL)	COVID-19-related social support service closures and mental well-being in older adults and those affected by dementia: A UK longitudinal survey
AUTHORS	Giebel, Clarissa; Pulford, Daniel; Cooper, Claudia; Lord, Kathryn; Shenton, Justine; Cannon, Jacqueline; Shaw, Lisa; Tetlow, Hilary; Limbert, Stan; Callaghan, Steve; Whittington, Rosie; Rogers, Carol; Komuravelli, Aravind; Rajagopal, Manoj; Eley, Ruth; Downs, Murna; Reilly, Siobhan; Ward, Kym; Gaughan, Anna; Butchard, Sarah; Beresford, Jules; Watkins, Caroline; Bennett, Kate; Gabbay, Mark

VERSION 1 – REVIEW

REVIEWER	Sanjeev Kumar University of Toronto, Canada
REVIEW RETURNED	29-Nov-2020

GENERAL COMMENTS	This study surveyed older adults, persons with dementia and their carers to assess social support service usage and symptoms of depression/anxiety/mental well being. This is a nice effort with a good sample size and the study is relevant to the current times and helps us understand the impact of pandemic on older adults. The manuscript is written well. I have following comments: 1. I think the major limitation of this study is the huge rate of drop-outs from the survey. Authors started with 660 participants and only 411 completed the T3 follow up. This might have significantly influenced the results and the people who dropped out may have been more likely to be not doing well (just my assumption, it might also be the other way around), have experienced challenges with the support services and may even be more likely to experience symptoms of depression/anxiety/stress. So, I believe some attention should be paid to the ones who dropped out as they are unlikely to be 'missing at Random'. There are several ways to analyse such data such as using likelihood models and pattern mixture models (https://www.jstor.org/stable/2345937, https://www.jstor.org/stable/43185208 and PMID: 8934587).2. As there were no a priori hypotheses, I am not sure how the authors set the significance at $p < 0.05$. Authors should make it clear that these were exploratory analyses and should probably control for multiple tests or acknowledge it as a limitation.3. It seems to me that authors have used the terms depression and anxiety to mean diagnostic entities of Depressive disorder or Anxiety disorders, which is not accurate. The scales such as PHQ - 9 and GAD-7 are validated measures to assess symptom burden but they can not establish diagnoses of these disorders. The diagnosis have to be confirmed with clinical interview or something like SCID. It might be better to use the terms symptoms. Moreover
---

	these symptoms in patients with dementia are likely to part of the cognitive disorder and these symptoms may have worsened during COVID (https://doi.org/10.3389/fpsyt.2020.573367). 4. Figure 1 is a bit hard to understand. Not sure what duplicates mean? how is missing different from lost to follow up? at T2, n = 481 and at T3 n = 411, but only 9 were lost to follow up? The numbers in figure one do not add up for T3. $37+147=42+148 = 377$? 5. Please correct typos etc. in references.
--	---

REVIEWER	Babak Tousi Cleveland Clinic, USA
REVIEW RETURNED	01-Dec-2020

GENERAL COMMENTS	It is more helpful if the authors elaborate if there was any correlation between the change of mood (depression or anxiety) and receiving or not receiving the social services post covid.  - The four subgroups and three groups have to be specifically defined. (i.e. the Former carer should be specified) - The use of group and subgroup should be used homogenously in the manuscript. It was difficult at times to know what group they were comparing to.
---

VERSION 1 – AUTHOR RESPONSE

Reviewer 1:

1. I think the major limitation of this study is the huge rate of drop-outs from the survey. Authors started with 660 participants and only 411 completed the T3 follow up. This might have significantly influenced the results and the people who dropped out may have been more likely to be not doing well (just my assumption, it might also be the other way around), have experienced challenges with the support services and may even be more likely to experience symptoms of depression/anxiety/stress. So, I believe some attention should be paid to the ones who dropped out as they are unlikely to be 'missing at Random'. There are several ways to analyse such data such as using likelihood models and pattern mixture models (<https://www.jstor.org/stable/2345937>, <https://www.jstor.org/stable/43185208> and PMID: 8934587).

→ We have now compared the levels of depression, anxiety, and well-being at T1 between those who subsequently dropped out and those who completed all three time points. No significant differences in means were found. We have added this now under Methods and in the Results.

2. As there were no a priori hypotheses, I am not sure how the authors set the significance at $p < 0.05$. Authors should make it clear that these were exploratory analyses and should probably control for multiple tests or acknowledge it as a limitation.

→ This exploratory study aimed to explore how social support service usage and mental health are longitudinally affected by the pandemic in dementia and ageing. To test of differences between the three survey time points, it is normal to use a p value to understand significance. We have now added that this is an exploratory study.

3. It seems to me that authors have used the terms depression and anxiety to mean diagnostic entities of Depressive disorder or Anxiety disorders, which is not accurate. The scales such as PHQ - 9 and GAD-7 are validated measures to assess symptom burden but they can not establish

diagnoses of these disorders. The diagnosis have to be confirmed with clinical interview or something like SCID. It might be better to use the terms symptoms. Moreover these symptoms in patients with dementia are likely to part of the cognitive disorder and these symptoms may have worsened during COVID (<https://doi.org/10.3389/fpsy.2020.573367>).

➔ Yes, a full diagnosis from a psychiatrist is needed to accurately state anxiety and depression. We have phrased this in the manuscript now that we are specific about that the GAD-7 and the PHQ-9 refer to levels of anxiety and depressive symptoms, and the cut offs are indicative of GAD and depression.

4. Figure 1 is a bit hard to understand. Not sure what duplicates mean? how is missing different from lost to follow up? at T2, n = 481 and at T3 n = 411, but only 9 were lost to follow up? The numbers in figure one do not add up for T3. $37+147=42+148 = 377?$

➔ Apologies, we have now checked the numbers and have corrected these, they now add up in the T3 box. We have also clarified further how the numbers are calculated in the Figure 1 legend.

5. Please correct typos etc. in references.

➔ We are unclear what the reviewer is referring to here. Having checked through the list of references, there are no typos. What is underscored in the word document are names which may appear as typos but are the relevant author names.

Reviewer: 2

It is more helpful if the authors elaborate if there was any correlation between the change of mood (depression or anxiety) and receiving or not receiving the social services post covid.

➔ There was no significant correlation between change in mood and receiving/not receiving social support services since COVID. We have now included this in the Methods and the Results.

- The four subgroups and three groups have to be specifically defined. (i.e. the Former carer should be specified)

➔ We have now specified this better under Participants and Recruitment.

- The use of group and subgroup should be used homogenously in the manuscript. It was difficult at times to know what group they were comparing to.

➔ Apologies, we have now removed all mentioning of subgroups and now referred throughout to groups.

VERSION 2 – REVIEW

REVIEWER	Sanjeev Kumar University of Toronto, Canada
REVIEW RETURNED	28-Dec-2020

GENERAL COMMENTS	Authors have failed to revise the manuscript and correct typos etc. as per recommendations of this reviewer. 1. Regarding the missing data, although authors conducted a baseline analysis and found no differences it does not mean that
--

	differences did not exist at follow up points, so such a large amount of missing data should have been acknowledged as a significant limitation, alternatively some other sophisticated analyses could be done as was suggested during the first review. Moreover for the T1 comparison authors have provided p values only showing that there were no differences. Isolated p values are not very meaningful in such cases. Another reason missing data is particularly problematic in this study is because authors are relying on proportion of patients with particular symptoms and presenting comparisons between the visits. 2. There are no hypotheses or specific objectives provided. 3. I could not find the revised participant flow diagram (figure 2) with revised manuscript and the numbers in the text are still unclear. Authors mention that only 377 participants were analysed as they completed all time points, however the statement is confusing “377 participants completed all three waves of the survey (37 PLWD; 147 current carers; 42 former carers; 148 older adults)”. The Total of numbers in brackets is 37+147+42+148 = 274. That further elevates the number of missing persons. 4. Authors continue to use terms depression and anxiety as pertaining to diagnoses of depressive disorder and generalized anxiety disorder based on their cut offs. As previously stated in the first review, PHQ-9 and GAD are not diagnostic instruments. Elevated scores on GAD-7 can not be taken as generalized anxiety disorder as anxiety symptoms could be due to adjustment disorder, part of depression syndrome or part of neuropsychiatric symptoms of dementia. Same is true for PHQ-9. This labeling is misleading. 5. Examples of typos: “Ethical approval was obtained from the University of Liverpool prior to study begin” “Brown EE, Kumr S, Rajji TK, et al. Anticipating and mitigating the impact of the COVID-19 pandemic on Alzheimer’s disease and related dementias. American Journal of Geriatric Psychiatry 2020;28(7):712-721.”
--	---

REVIEWER	Babak Tousi MD Cleveland Clinic , USA
REVIEW RETURNED	23-Dec-2020

GENERAL COMMENTS	That was relevant topic in a difficult time. It may help to prioritize the resources down the road when pandemic have a bigger toll on the society.
---

VERSION 2 – AUTHOR RESPONSE

Reviewer 1:

1. Regarding the missing data, although authors conducted a baseline analysis and found no differences it does not mean that differences did not exist at follow up points, so such a large amount of missing data should have been acknowledged as a significant limitation, alternatively some other sophisticated analyses could be done as was suggested during the first review. Moreover for the T1 comparison authors have provided p values only showing that there were no differences. Isolated p values are not very meaningful in such cases. Another reason missing data is particularly problematic in this study is because authors are relying on proportion of patients with particular symptoms and presenting comparisons between the visits.

→ In our previous revision, as suggested, we had already compared the mental health scores of those who completed all three survey time points with those who dropped out after T1 or T2, showing no significant differences in mean scores. Longitudinal survey data always contains missing data, particularly with longer surveys, yet we still managed successfully to obtain data from 377 participants across the three time points, which is a strength of this novel study. As the reviewer recommends, we have added the missing data into the limitations section.

2. There are no hypotheses or specific objectives provided.

→ The reviewer has not highlighted this previously, and this appears to be a new comment. Therefore, we would not have been able to address this until now. We have now specified this.

3. I could not find the revised participant flow diagram (figure 2) with revised manuscript and the numbers in the text are still unclear. Authors mention that only 377 participants were analysed as they completed all time points, however the statement is confusing “377 participants completed all three waves of the survey (37 PLWD; 147 current carers; 42 former carers; 148 older adults)”. The Total of numbers in brackets is $37+147+42+148 = 274$. That further elevates the number of missing persons.

→ This had been updated and added (see Figure 1, note not Figure 2), and its updated legend. We have also clarified the number in the text.

4. Authors continue to use terms depression and anxiety as pertaining to diagnoses of depressive disorder and generalized anxiety disorder based on their cut offs. As previously stated in the first review, PHQ-9 and GAD are not diagnostic instruments. Elevated scores on GAD-7 can not be taken as generalized anxiety disorder as anxiety symptoms could be due to adjustment disorder, part of depression syndrome or part of neuropsychiatric symptoms of dementia. Same is true for PHQ-9. This labeling is misleading.

→ This terminology is being used in many other publications, and as stated, we are referring to the terminology of the GAD-7 and the PHQ-9. As outlined in our previous revisions, we have clearly stated in the manuscript that the GAD-7 and the PHQ-9 are referring to levels of anxiety and depression, and those that have reached the cut off or above are indicative of anxiety and depression, as tested and found reliable and valid in GAD-7 and PHQ-9 based research. We have gone through the document again to ensure this is clearer, but this is what the research is labelling PHQ-9 and GAD-7, and we are merely referring to this.

5. Examples of typos: “Ethical approval was obtained from the University of Liverpool prior to study begin”

“Brown EE, Kumr S, Rajji TK, et al. Anticipating and mitigating the impact of the COVID-19 pandemic on Alzheimer’s disease and related dementias. American Journal of Geriatric Psychiatry 2020;28(7):712-721.”

→ There is no typo in the sentence on ethical approval. Equally, there is no typo in the reference. We do not understand what the reviewer is referring to, as there are no typos.

Reviewer: 2

No amendments required.